# Mutual information for symmetric rank-one matrix estimation: A proof of the replica formula

**Jean Barbier, Mohamad Dia and Nicolas Macris**
Laboratoire de Théorie des Communications, Faculté Informatique et Communications,
Ecole Polytechnique Fédérale de Lausanne, 1015, Suisse.
`firstname.lastname@epfl.ch`

**Florent Krzakala**
Laboratoire de Physique Statistique, CNRS, PSL Universités et Ecole Normale Supérieure,
Sorbonne Universités et Université Pierre & Marie Curie, 75005, Paris, France.
`florent.krzakala@ens.fr`

**Thibault Lesieur and Lenka Zdeborová**
Institut de Physique Théorique, CNRS, CEA, Université Paris-Saclay, F-91191, Gif-sur-Yvette, France.
`lesieur.thibault@gmail.com,lenka.zdeborova@gmail.com`

## Abstract

Factorizing low-rank matrices has many applications in machine learning and statistics. For probabilistic models in the Bayes optimal setting, a general expression for the mutual information has been proposed using heuristic statistical physics computations, and proven in few specific cases. Here, we show how to rigorously prove the conjectured formula for the symmetric rank-one case. This allows to express the minimal mean-square-error and to characterize the detectability phase transitions in a large set of estimation problems ranging from community detection to sparse PCA. We also show that for a large set of parameters, an iterative algorithm called approximate message-passing is Bayes optimal. There exists, however, a gap between what currently known polynomial algorithms can do and what is expected information theoretically. Additionally, the proof technique has an interest of its own and exploits three essential ingredients: the interpolation method introduced in statistical physics by Guerra, the analysis of the approximate message-passing algorithm and the theory of spatial coupling and threshold saturation in coding. Our approach is generic and applicable to other open problems in statistical estimation where heuristic statistical physics predictions are available.

Consider the following probabilistic rank-one matrix estimation problem: one has access to noisy observations $\mathbf{w} = (w_{ij})_{i,j=1}^{n}$ of the pair-wise product of the components of a vector $\mathbf{s} = (s_1, \ldots, s_n)^{\mathsf{T}} \in \mathbb{R}^n$ with i.i.d components distributed as $S_i \sim P_0$, $i = 1, \ldots, n$. The entries of $\mathbf{w}$ are observed through a noisy element-wise (possibly non-linear) output probabilistic channel $P_{\text{out}}(w_{ij}|s_i s_j/\sqrt{n})$. The goal is to estimate the vector $\mathbf{s}$ from $\mathbf{w}$ assuming that both $P_0$ and $P_{\text{out}}$ are known and independent of $n$ (noise is symmetric so that $w_{ij} = w_{ji}$). Many important problems in statistics and machine learning can be expressed in this way, such as sparse PCA [1], the Wigner spiked model [2, 3], community detection [4] or matrix completion [5].

Proving a result initially derived by a heuristic method from statistical physics, we give an explicit expression for the mutual information (MI) and the information theoretic minimal mean-square-error (MMSE) in the asymptotic $n \to \infty$ limit. Our results imply that for a large region of parameters, the posterior marginal expectations of the underlying signal components (often assumed intractable

to compute) can be obtained in the leading order in $n$ using a polynomial-time algorithm called approximate message-passing (AMP) [6, 3, 4, 7]. We also demonstrate the existence of a region where both AMP and spectral methods [8] fail to provide a good answer to the estimation problem, while it is nevertheless information theoretically possible to do so. We illustrate our theorems with examples and also briefly discuss the implications in terms of computational complexity.

# 1 Setting and main results

**The additive white Gaussian noise setting:** A standard and natural setting is the case of additive white Gaussian noise (AWGN) of known variance $\Delta$, $w_{ij} = s_i s_j / \sqrt{n} + z_{ij} \sqrt{\Delta}$, where $\mathbf{z} = (z_{ij})_{i,j=1}^n$ is a symmetric matrix with i.i.d entries $Z_{ij} \sim \mathcal{N}(0,1)$, $1 \le i \le j \le n$. Perhaps surprisingly, it turns out that this Gaussian setting is sufficient to completely characterize all the problems discussed in the introduction, even if these have more complicated output channels. This is made possible by a theorem of channel universality [9] (already proven for community detection in [4] and conjectured in [10]). This theorem states that given an output channel $P_{\text{out}}(w|y)$, such that (s.t) $\log P_{\text{out}}(w|y=0)$ is three times differentiable with bounded second and third derivatives, then the MI satisfies $I(\mathbf{S}; \mathbf{W}) = I(\mathbf{S}; \mathbf{SS}^\mathsf{T} / \sqrt{n} + \mathbf{Z}\sqrt{\Delta}) + \mathcal{O}(\sqrt{n})$, where $\Delta$ is the inverse Fisher information (evaluated at $y=0$) of the output channel: $\Delta^{-1} := \mathbb{E}_{P_{\text{out}}(w|0)}[(\partial_y \log P_{\text{out}}(W|y)|_{y=0})^2]$. Informally, this means that we only have to compute the MI for an AWGN channel to take care of a wide range of problems, which can be expressed in terms of their Fisher information. In this paper we derive rigorously, for a large class of signal distributions $P_0$, an explicit one-letter formula for the MI per variable $I(\mathbf{S}; \mathbf{W})/n$ in the asymptotic limit $n \to \infty$.

**Main result:** Our central result is a proof of the expression for the asymptotic $n \to \infty$ MI per variable via the so-called *replica symmetric (RS) potential* $i_{\text{RS}}(E; \Delta)$ defined as

$$i_{\text{RS}}(E; \Delta) := \frac{(v-E)^2 + v^2}{4\Delta} - \mathbb{E}_{S,Z}\left[\ln\left(\int dx\, P_0(x) e^{-\frac{x^2}{2\Sigma(E;\Delta)^2} + x\left(\frac{S}{\Sigma(E;\Delta)^2} + \frac{Z}{\Sigma(E;\Delta)}\right)}\right)\right], \quad (1)$$

with $Z \sim \mathcal{N}(0,1)$, $S \sim P_0$, $\mathbb{E}[S^2] = v$ and $\Sigma(E;\Delta)^2 := \Delta/(v-E)$, $E \in [0,v]$. Here we will assume that $P_0$ is a discrete distribution over a finite bounded real alphabet $P_0(s) = \sum_{\alpha=1}^\nu p_\alpha \delta(s-a_\alpha)$. Thus the only continuous integral in (1) is the Gaussian over $z$. Our results can be extended to mixtures of discrete and continuous signal distributions at the expense of technical complications in some proofs.

It turns out that both the information theoretic and algorithmic AMP thresholds are determined by the set of stationary points of (1) (w.r.t $E$). It is possible to show that for all $\Delta > 0$ there always exist at least one stationary minimum. Note $E=0$ is never a stationary point (except for $P_0$ a single Dirac mass) and $E=v$ is stationary only if $\mathbb{E}[S]=0$. In this contribution we suppose that at most three stationary points exist, corresponding to situations with at most one phase transition. We believe that situations with multiple transitions can also be covered by our techniques.

**Theorem 1.1 (RS formula for the mutual information)** *Fix $\Delta > 0$ and let $P_0$ be a discrete distribution s.t (1) has at most three stationary points. Then $\lim_{n\to\infty} I(\mathbf{S}; \mathbf{W})/n = \min_{E \in [0,v]} i_{\text{RS}}(E; \Delta)$.*

The proof of the *existence of the limit* does not require the above hypothesis on $P_0$. Also, it was first shown in [9] that for all $n$, $I(\mathbf{S}; \mathbf{W})/n \le \min_{E \in [0,v]} i_{\text{RS}}(E; \Delta)$, an inequality that *we will use* in the proof section. It is conceptually useful to define the following threshold:

**Definition 1.2 (Information theoretic threshold)** *Define $\Delta_{\text{Opt}}$ as the first non-analyticity point of the MI as $\Delta$ increases: $\Delta_{\text{Opt}} := \sup\{\Delta \,|\, \lim_{n\to\infty} I(\mathbf{S}; \mathbf{W})/n \text{ is analytic in } ]0, \Delta[\}$.*

When $P_0$ is s.t (1) has at most three stationary points, as discussed below, then $\min_{E \in [0,v]} i_{\text{RS}}(E; \Delta)$ has at most one non-analyticity point denoted $\Delta_{\text{RS}}$ (if $\min_{E \in [0,v]} i_{\text{RS}}(E; \Delta)$ is analytic over all $\mathbb{R}_+$ we set $\Delta_{\text{RS}} = \infty$). Theorem 1.1 gives us a mean to *compute* the information theoretic threshold $\Delta_{\text{Opt}} = \Delta_{\text{RS}}$. A basic application of theorem 1.1 is the expression of the MMSE:

**Corollary 1.3 (Exact formula for the MMSE)** *For all $\Delta \neq \Delta_{\text{RS}}$, the matrix-MMSE $\mathrm{Mmmse}_n := \mathbb{E}_{S,W}[\|\mathbf{SS}^\mathsf{T} - \mathbb{E}[\mathbf{XX}^\mathsf{T}|\mathbf{W}]\|_{\mathrm{F}}^2]/n^2$ ($\|-\|_{\mathrm{F}}$ being the Frobenius norm) is asymptotically $\lim_{n\to\infty} \mathrm{Mmmse}_n(\Delta^{-1}) = v^2 - (v - \mathrm{argmin}_{E \in [0,v]} i_{\text{RS}}(E; \Delta))^2$. Moreover, if $\Delta < \Delta_{\text{AMP}}$ (where $\Delta_{\text{AMP}}$ is the algorithmic threshold, see definition 1.4) or $\Delta > \Delta_{\text{RS}}$, then the usual vector-MMSE $\mathrm{Vmmse}_n := \mathbb{E}_{S,W}[\|\mathbf{S} - \mathbb{E}[\mathbf{X}|\mathbf{W}]\|_2^2]/n$ satisfies $\lim_{n\to\infty} \mathrm{Vmmse}_n = \mathrm{argmin}_{E \in [0,v]} i_{\text{RS}}(E; \Delta)$.*

It is natural to conjecture that the vector-MMSE is given by $\mathrm{argmin}_{E \in [0,v]} i_{\mathrm{RS}}(E; \Delta)$ for all $\Delta \neq \Delta_{\mathrm{RS}}$, but our proof does not quite yield the full statement.

A fundamental consequence concerns the performance of the AMP algorithm [6] for estimating $\mathbf{s}$. AMP has been analysed rigorously in [11, 12, 4] where it is shown that its asymptotic performance is tracked by *state evolution* (SE). Let $E^t := \lim_{n \to \infty} \mathbb{E}_{\mathbf{S}, \mathbf{Z}}[\|\mathbf{S} - \hat{\mathbf{s}}^t\|_2^2]/n$ be the asymptotic average vector-MSE of the AMP estimate $\hat{\mathbf{s}}^t$ at time $t$. Define $\mathrm{mmse}(\Sigma^{-2}) := \mathbb{E}_{S,Z}[(S - \mathbb{E}[X|S + \Sigma Z])^2]$ as the usual scalar mmse function associated to a scalar AWGN channel of noise variance $\Sigma^2$, with $S \sim P_0$ and $Z \sim \mathcal{N}(0,1)$. Then

$$E^{t+1} = \mathrm{mmse}(\Sigma(E^t; \Delta)^{-2}), \qquad E^0 = v, \tag{2}$$

is the SE recursion. Monotonicity properties of the mmse function imply that $E^t$ is a decreasing sequence s.t $\lim_{t \to \infty} E^t = E^\infty$ exists. Note that when $\mathbb{E}[S] = 0$ and $v$ is an unstable fixed point, as such, SE "does not start". While this is not really a problem when one runs AMP in practice, for analysis purposes one can slightly bias $P_0$ and remove the bias at the end of the proofs.

**Definition 1.4 (AMP algorithmic threshold)** *For $\Delta > 0$ small enough, the fixed point equation corresponding to* (2) *has a unique solution for all noise values in* $]0, \Delta[$. *We define $\Delta_{\mathrm{AMP}}$ as the supremum of all such $\Delta$.*

**Corollary 1.5 (Performance of AMP)** *In the limit $n \to \infty$, AMP initialized without any knowledge other than $P_0$ yields upon convergence the asymptotic matrix-MMSE as well as the asymptotic vector-MMSE iff $\Delta < \Delta_{\mathrm{AMP}}$ or $\Delta > \Delta_{\mathrm{RS}}$, namely $E^\infty = argmin_{E \in [0,v]} i_{\mathrm{RS}}(E; \Delta)$.*

$\Delta_{\mathrm{AMP}}$ can be read off the replica potential (1): by differentiation of (1) one finds a fixed point equation that corresponds to (2). Thus $\Delta_{\mathrm{AMP}}$ is the smallest solution of $\partial i_{\mathrm{RS}}/\partial E = \partial^2 i_{\mathrm{RS}}/\partial E^2 = 0$; in other words it is the "first" horizontal inflexion point appearing in $i_{\mathrm{RS}}(E; \Delta)$ when $\Delta$ increases.

**Discussion:** With our hypothesis on $P_0$ there are only three possible scenarios: $\Delta_{\mathrm{AMP}} < \Delta_{\mathrm{RS}}$ (one "first order" phase transition); $\Delta_{\mathrm{AMP}} = \Delta_{\mathrm{RS}} < \infty$ (one "higher order" phase transition); $\Delta_{\mathrm{AMP}} = \Delta_{\mathrm{RS}} = \infty$ (no phase transition). In the sequel we will have in mind the most interesting case, namely *one first order phase transition*, where we determine the gap between the algorithmic AMP and information theoretic performance. The cases of no phase transition or higher order phase transition, which present no algorithmic gap, are basically covered by the analysis of [3] and follow as a special case from our proof. The only cases that would require more work are those where $P_0$ is s.t (1) develops more than three stationary points and more than one phase transition is present.

For $\Delta_{\mathrm{AMP}} < \Delta_{\mathrm{RS}}$ the structure of stationary points of (1) is as follows[1] (figure 1). There exist three branches $E_{\mathrm{good}}(\Delta)$, $E_{\mathrm{unstable}}(\Delta)$ and $E_{\mathrm{bad}}(\Delta)$ s.t: **1)** For $0 < \Delta < \Delta_{\mathrm{AMP}}$ there is a single stationary point $E_{\mathrm{good}}(\Delta)$ which is a global minimum; **2)** At $\Delta_{\mathrm{AMP}}$ a *horizontal inflexion point* appears, for $\Delta \in [\Delta_{\mathrm{AMP}}, \Delta_{\mathrm{RS}}]$ there are three stationary points satisfying $E_{\mathrm{good}}(\Delta_{\mathrm{AMP}}) < E_{\mathrm{unstable}}(\Delta_{\mathrm{AMP}}) = E_{\mathrm{bad}}(\Delta_{\mathrm{AMP}})$, $E_{\mathrm{good}}(\Delta) < E_{\mathrm{unstable}}(\Delta) < E_{\mathrm{bad}}(\Delta)$ otherwise, and moreover $i_{\mathrm{RS}}(E_{\mathrm{good}}; \Delta) \leq i_{\mathrm{RS}}(E_{\mathrm{bad}}; \Delta)$ *with equality only at* $\Delta_{\mathrm{RS}}$; **3)** for $\Delta > \Delta_{\mathrm{RS}}$ there is *at least* the stationary point $E_{\mathrm{bad}}(\Delta)$ which is always the global minimum, i.e. $i_{\mathrm{RS}}(E_{\mathrm{bad}}; \Delta) < i_{\mathrm{RS}}(E_{\mathrm{good}}; \Delta)$. (For higher $\Delta$ the $E_{\mathrm{good}}(\Delta)$ and $E_{\mathrm{unstable}}(\Delta)$ branches may merge and disappear); **4)** $E_{\mathrm{good}}(\Delta)$ is analytic for $\Delta \in ]0, \Delta'[$, $\Delta' > \Delta_{\mathrm{RS}}$, and $E_{\mathrm{bad}}(\Delta)$ is analytic for $\Delta > \Delta_{\mathrm{AMP}}$.

We note for further use in the proof section that $E^\infty = E_{\mathrm{good}}(\Delta)$ for $\Delta < \Delta_{\mathrm{AMP}}$ and $E^\infty = E_{\mathrm{bad}}(\Delta)$ for $\Delta > \Delta_{\mathrm{AMP}}$. Definition 1.4 is equivalent to $\Delta_{\mathrm{AMP}} = \sup\{\Delta | E^\infty = E_{\mathrm{good}}(\Delta)\}$. Moreover we will also use that $i_{\mathrm{RS}}(E_{\mathrm{good}}; \Delta)$ is analytic on $]0, \Delta'[$, $i_{\mathrm{RS}}(E_{\mathrm{bad}}; \Delta)$ is analytic on $]\Delta_{\mathrm{AMP}}, \infty[$, and the only non-analyticity point of $\min_{E \in [0,v]} i_{\mathrm{RS}}(E; \Delta)$ is at $\Delta_{\mathrm{RS}}$.

**Relation to other works:** Explicit single-letter characterization of the MI in the rank-one problem has attracted a lot of attention recently. Particular cases of theorem 1.1 have been shown rigorously in a number of situations. A special case when $s_i = \pm 1 \sim \mathrm{Ber}(1/2)$ already appeared in [13] where an equivalent spin glass model is analysed. Very recently, [9] has generalized the results of [13] and, notably, obtained a generic matching upper bound. The same formula has been also rigorously computed following the study of AMP in [3] for spiked models (provided, however, that the signal was not *too* sparse) and in [4] for strictly symmetric community detection.

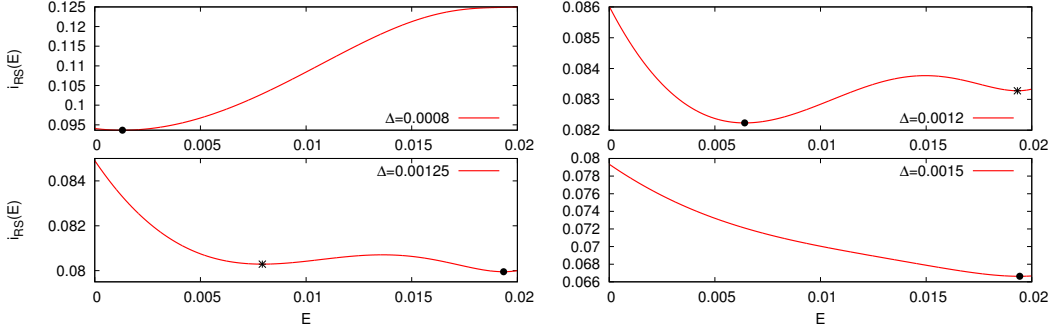

Figure 1: The replica symmetric potential $i_{\rm RS}(E)$ for four values of $\Delta$ in the Wigner spiked model. The MI is $\min i_{\rm RS}(E)$ (the black dot, while the black cross corresponds to the local minimum) and the asymptotic matrix-MMSE is $v^2 - (v - \mathrm{argmin}_E i_{\rm RS}(E))^2$, where $v = \rho$ in this case with $\rho = 0.02$ as in the inset of figure 2. From top left to bottom right: **(1)** For low noise values, here $\Delta = 0.0008 < \Delta_{\rm AMP}$, there exists a unique "good" minimum corresponding to the MMSE and AMP is Bayes optimal. **(2)** As the noise increases, a second local "bad" minimum appears: this is the situation at $\Delta_{\rm AMP} < \Delta = 0.0012 < \Delta_{\rm RS}$. **(3)** For $\Delta = 0.00125 > \Delta_{\rm RS}$, the "bad" minimum becomes the global one and the MMSE suddenly deteriorates. **(4)** For larger values of $\Delta$, only the "bad" minimum exists. AMP can be seen as a naive minimizer of this curve starting from $E = v = 0.02$. It reaches the global minimum in situations (1), (3) and (4), but in (2), when $\Delta_{\rm AMP} < \Delta < \Delta_{\rm RS}$, it is trapped by the local minimum with large MSE instead of reaching the global one corresponding to the MMSE.

For rank-one symmetric matrix estimation problems, AMP has been introduced by [6], who also computed the SE formula to analyse its performance, generalizing techniques developed by [11] and [12]. SE was further studied by [3] and [4]. In [7, 10], the generalization to larger rank was also considered. The general formula proposed by [10] for the conditional entropy and the MMSE on the basis of the heuristic cavity method from statistical physics was not demonstrated in full generality. Worst, all existing proofs could not reach the more interesting regime where a gap between the algorithmic and information theoretic perfomances appears, leaving a gap with the statistical physics conjectured formula (and rigorous upper bound from [9]). Our result closes this conjecture and has interesting non-trivial implications on the computational complexity of these tasks.

Our proof technique combines recent rigorous results in coding theory along the study of capacity-achieving spatially coupled codes [14, 15, 16, 17] with other progress, coming from developments in mathematical physics putting on a rigorous basis predictions of spin glass theory [18]. From this point of view, the theorem proved in this paper is relevant in a broader context going beyond low-rank matrix estimation. Hundreds of papers have been published in statistics, machine learning or information theory using the non-rigorous statistical physics approach. We believe that our result helps setting a rigorous foundation of a broad line of work. While we focus on rank-one symmetric matrix estimation, our proof technique is readily extendable to more generic low-rank symmetric matrix or low-rank symmetric tensor estimation. We also believe that it can be extended to other problems of interest in machine learning and signal processing, such as generalized linear regression, features/dictionary learning, compressed sensing or multi-layer neural networks.

## 2 Two examples: Wigner spiked model and community detection

In order to illustrate the consequences of our results we shall present two examples.

**Wigner spiked model:** In this model, the vector **s** is a Bernoulli random vector, $S_i \sim \mathrm{Ber}(\rho)$. For large enough densities (i.e. $\rho > 0.041(1)$), [3] computed the matrix-MMSE and proved that AMP is a computationally efficient algorithm that asymptotically achieves the matrix-MMSE for *any* value of the noise $\Delta$. Our results allow to close the gap left open by [3]: on one hand we now obtain rigorously the MMSE for $\rho \leq 0.041(1)$, and on the other one we observe that for such values of $\rho$, and as $\Delta$ decreases, there is a small region where two local minima coexist in $i_{\rm RS}(E; \Delta)$. In particular for $\Delta_{\rm AMP} < \Delta < \Delta_{\rm Opt} = \Delta_{\rm RS}$ the global minimum corresponding to the MMSE differs from the local one that traps AMP, and a computational gap appears (see figure 1). While the region where AMP is Bayes optimal is quite large, the region where is it not, however, is perhaps the most interesting one. While this is by no means evident, statistical physics analogies with physical phase transitions in nature suggest that this region should be hard for a very broad class of algorithms. For small $\rho$ our

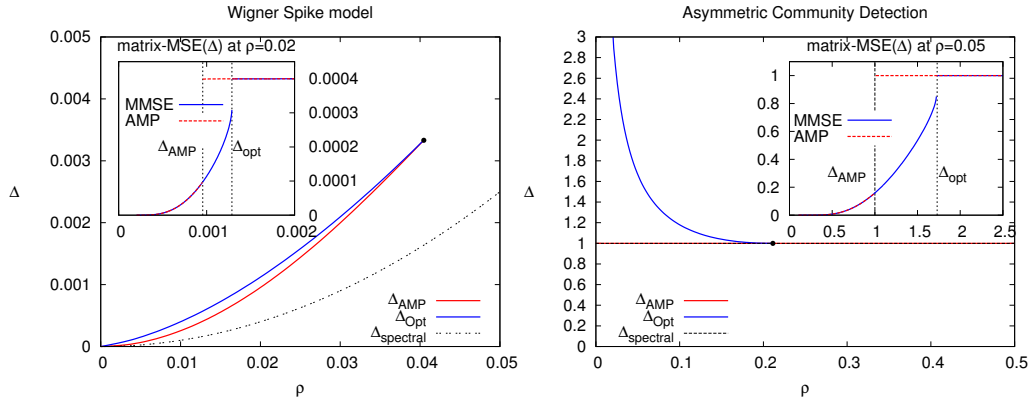

Figure 2: Phase diagram in the noise variance $\Delta$ versus density $\rho$ plane for the rank-one spiked Wigner model (left) and the asymmetric community detection (right). **Left:** [3] proved that AMP achieves the matrix-MMSE for all $\Delta$ as long as $\rho > 0.041(1)$. Here we show that AMP is actually achieving the optimal reconstruction in the whole phase diagram except in the small region between the blue and red lines. Notice the large gap with spectral methods (dashed black line). **Inset:** matrix-MMSE (blue) at $\rho = 0.02$ as a function of $\Delta$. AMP (dashed red) provably achieves the matrix-MMSE except in the region $\Delta_{\mathrm{AMP}} < \Delta < \Delta_{\mathrm{Opt}} = \Delta_{\mathrm{RS}}$. We conjecture that no polynomial-time algorithm will do better than AMP in this region. **Right:** Asymmetric community detection problem with two communities. For $\rho > 1/2 - \sqrt{1/12}$ (black point) and when $\Delta > 1$, it is information theoretically impossible to find any overlap with the true communities and the matrix-MMSE is 1, while it becomes possible for $\Delta < 1$. In this region, AMP is always achieving the matrix-MMSE and spectral methods can find a non-trivial overlap with the truth as well, starting from $\Delta < 1$. For $\rho < 1/2 - \sqrt{1/12}$, however, it is information theoretically possible to find an overlap with the hidden communities for $\Delta > 1$ (below the blue line) but both AMP and spectral methods miss this information. **Inset:** matrix-MMSE (blue) at $\rho = 0.05$ as a function of $\Delta$. AMP (dashed red) again provably achieves the matrix-MMSE except in the region $\Delta_{\mathrm{AMP}} < \Delta < \Delta_{\mathrm{Opt}}$.

results are consistent with the known optimal and algorithmic thresholds predicted in sparse PCA [19, 20], that treats the case of sub-extensive $\rho = \mathcal{O}(1)$ values. Another interesting line of work for such probabilistic models appeared in the context of random matrix theory (see [8] and references therein) and predicts that a sharp phase transition occurs at a critical value of the noise $\Delta_{\mathrm{spectral}} = \rho^2$ below which an outlier eigenvalue (and its principal eigenvector) has a positive correlation with the hidden signal. For larger noise values the spectral distribution of the observation is indistinguishable from that of the pure random noise.

**Asymmetric balanced community detection:** We now consider the problem of detecting two communities (groups) with different sizes $\rho n$ and $(1-\rho)n$, that generalizes the one considered in [4]. One is given a graph where the probability to have a link between nodes in the first group is $p + \mu(1-\rho)/(\rho\sqrt{n})$, between those in the second group is $p + \mu\rho/(\sqrt{n}(1-\rho))$, while inter-connections appear with probability $p - \mu/\sqrt{n}$. With this peculiar "balanced" setting, the nodes in each group have the same degree distribution with mean $pn$, making them harder to distinguish. According to the universality property described in the first section, this is equivalent to a model with AWGN of variance $\Delta = p(1-p)/\mu^2$ where each variable $s_i$ is chosen according to $P_0(s) = \rho\delta(s - \sqrt{(1-\rho)/\rho}) + (1-\rho)\delta(s + \sqrt{\rho/(1-\rho)})$. Our results for this problem[2] are summarized on the right hand side of figure 2. For $\rho > \rho_c = 1/2 - \sqrt{1/12}$ (black point), it is asymptotically information theoretically *possible* to get an estimation better than chance if and only if $\Delta < 1$. When $\rho < \rho_c$, however, it becomes possible for much larger values of the noise. Interestingly, AMP and spectral methods have the same transition and can find a positive correlation with the hidden communities for $\Delta < 1$, regardless of the value of $\rho$. Again, a region $[\Delta_{\mathrm{AMP}}, \Delta_{\mathrm{Opt}} = \Delta_{\mathrm{RS}}]$ exists where a computational gap appears when $\rho < \rho_c$. One can investigate the very low $\rho$ regime where we find that the information theoretic transition goes as $\Delta_{\mathrm{Opt}}(\rho \to 0) = 1/(4\rho|\log \rho|)$. Now if we assume that this result stays true even for $\rho = \mathcal{O}(1)$ (which is a speculation at this point), we can choose $\mu \to (1-p)\rho\sqrt{n}$ such that the small group is a clique. Then the problem corresponds to a "balanced" version of the famous planted clique problem [21]. We find that the AMP/spectral approach finds the

hidden clique when it is larger than $\sqrt{np/(1-p)}$, while the information theoretic transition translates into size of the clique $4p\log(n)/(1-p)$. This is indeed reminiscent of the more classical planted clique problem at $p=1/2$ with its gap between $\log(n)$ (information theoretic), $\sqrt{n/e}$ (AMP [22]) and $\sqrt{n}$ (spectral [21]). Since in our balanced case the spectral and AMP limits match, this suggests that the small gain of AMP in the standard clique problem is simply due to the information provided by the distribution of local degrees in the two groups (which is absent in our balanced case). We believe this correspondence strengthens the claim that the AMP gap is actually a fundamental one.

## 3 Proofs

The crux of our proof rests on an auxiliary "spatially coupled system". The hallmark of spatially coupled models is that one can tune them so that the gap between the algorithmic and information theoretic limits is eliminated, while at the same time the MI is maintained unchanged for the coupled and original models. Roughly speaking, this means that it is possible to algorithmically compute the information theoretic limit of the original model because a suitable algorithm is optimal on the coupled system. The present spatially coupled construction is similar to the one used for the coupled Curie-Weiss model [14]. Consider a ring of length $L+1$ ($L$ even) with *blocks* positioned at $\mu \in \{0, \dots, L\}$ and coupled to neighboring blocks $\{\mu-w, \dots, \mu+w\}$. Positions $\mu$ are taken modulo $L+1$ and the integer $w \in \{0, \dots, L/2\}$ equals the size of the *coupling window*. The coupled model is

$$w_{i_\mu j_\nu} = s_{i_\mu} s_{j_\nu} \sqrt{\frac{\Lambda_{\mu\nu}}{n}} + z_{i_\mu j_\nu} \sqrt{\Delta}, \qquad (3)$$

where the index $i_\mu \in \{1, \dots, n\}$ (resp. $j_\nu$) belongs to the block $\mu$ (resp. $\nu$) along the ring, $\mathbf{\Lambda}$ is an $(L+1) \times (L+1)$ matrix which describes the strength of the coupling between blocks, and $Z_{i_\mu j_\nu} \sim \mathcal{N}(0,1)$ are i.i.d. For the proof to work, the matrix elements have to be chosen appropriately. We assume that: $i)$ $\mathbf{\Lambda}$ is a doubly stochastic matrix; $ii)$ $\Lambda_{\mu\nu}$ depends on $|\mu-\nu|$; $iii)$ $\Lambda_{\mu\nu}$ is not vanishing for $|\mu-\nu| \leq w$ and vanishes for $|\mu-\nu| > w$; $iv)$ $\mathbf{\Lambda}$ is *smooth* in the sense $|\Lambda_{\mu\nu} - \Lambda_{\mu+1\nu}| = \mathcal{O}(w^{-2})$; $v)$ $\mathbf{\Lambda}$ has a non-negative Fourier transform. All these conditions can easily be met, the simplest example being a triangle of base $2w+1$ and height $1/(w+1)$. The construction of the coupled system is completed by introducing a *seed* in the ring: we assume perfect knowledge of the signal components $\{s_{i_\mu}\}$ for $\mu \in \mathcal{B} := \{-w-1, \dots, w-1\} \mod L+1$. This seed is what allows to close the gap between the algorithmic and information theoretic limits and therefore plays a crucial role. Note it can also be viewed as an "opening" of the chain with fixed boundary conditions. Our first crucial result states that the MI $I_{w,L}(\mathbf{S}; \mathbf{W})$ of the coupled and original systems are the same in a suitable limit.

**Lemma 3.1 (Equality of mutual informations)** *For any fixed $w$ the following limits exist and are equal:* $\lim_{L\to\infty} \lim_{n\to\infty} I_{w,L}(\mathbf{S}; \mathbf{W})/(n(L+1)) = \lim_{n\to\infty} I(\mathbf{S}; \mathbf{W})/n$.

An immediate corollary is that non-analyticity points (w.r.t $\Delta$) of the MIs are the same in the coupled and original models. In particular, defining $\Delta_{\mathrm{Opt,coup}} := \sup\{\Delta \mid \lim_{L\to\infty} \lim_{n\to\infty} I_{w,L}(\mathbf{S}; \mathbf{W})/(n(L+1))$ is analytic in $]0, \Delta[\}$, we have $\Delta_{\mathrm{Opt,coup}} = \Delta_{\mathrm{Opt}}$.

The second crucial result states that the AMP threshold of the spatially coupled system is at least as good as $\Delta_{\mathrm{RS}}$. The analysis of AMP applies to the coupled system as well [11, 12] and it can be shown that the performance of AMP is assessed by SE. Let $E_\mu^t := \lim_{n\to\infty} \mathbb{E}_{\mathbf{S},\mathbf{z}}[\|\mathbf{S}_\mu - \hat{\mathbf{s}}_\mu^t\|_2^2]/n$ be the asymptotic average vector-MSE of the AMP estimate $\hat{\mathbf{s}}_\mu^t$ at time $t$ for the $\mu$-th "block" of $\mathbf{S}$. We associate to each position $\mu \in \{0, \dots, L\}$ an *independent* scalar system with AWGN of the form $Y = S + \Sigma_\mu(\mathbf{E}; \Delta)Z$, with $\Sigma_\mu(\mathbf{E}; \Delta)^2 := \Delta/(v - \sum_{\nu=0}^L \Lambda_{\mu\nu} E_\nu)$ and $S \sim P_0$, $Z \sim \mathcal{N}(0,1)$. Taking into account knowledge of the signal components in $\mathcal{B}$, SE reads:

$$E_\mu^{t+1} = \mathrm{mmse}(\Sigma_\mu(\mathbf{E}^t; \Delta)^{-2}), \ E_\mu^0 = v \text{ for } \mu \in \{0, \dots, L\} \setminus \mathcal{B}, \ E_\mu^t = 0 \text{ for } \mu \in \mathcal{B}, t \geq 0, \quad (4)$$

where the mmse function is defined as in section 1. From the monotonicity of the mmse function we have $E_\mu^{t+1} \leq E_\mu^t$ for all $\mu \in \{0, \dots, L\}$, a partial order which implies that $\lim_{t\to\infty} \mathbf{E}^t = \mathbf{E}^\infty$ exists. This allows to define an algorithmic threshold for the coupled system: $\Delta_{\mathrm{AMP},w,L} := \sup\{\Delta | E_\mu^\infty \leq E_{\mathrm{good}}(\Delta) \ \forall \ \mu\}$. We show (equality holds but is not directly needed):

**Lemma 3.2 (Threshold saturation)** *Let* $\Delta_{\mathrm{AMP,coup}} := \liminf_{w\to\infty} \liminf_{L\to\infty} \Delta_{\mathrm{AMP},w,L}$. *We have* $\Delta_{\mathrm{AMP,coup}} \geq \Delta_{\mathrm{RS}}$.

**Proof sketch of theorem 1.1:** *First we prove the RS formula for $\Delta \leq \Delta_{\mathrm{Opt}}$.* It is known [3] that the matrix-MSE of AMP when $n \to \infty$ is equal to $v^2 - (v - E^t)^2$. This cannot improve the matrix-MMSE, hence $(v^2 - (v - E^\infty)^2)/4 \geq \limsup_{n\to\infty} \mathrm{Mmmse}_n/4$. For $\Delta \leq \Delta_{\mathrm{AMP}}$ we have $E^\infty = E_{\mathrm{good}}(\Delta)$ which is the global minimum of (1) so the left hand side of the last inequality equals the derivative of $\min_{E\in[0,v]} i_{\mathrm{RS}}(E; \Delta)$ w.r.t $\Delta^{-1}$. Thus using the matrix version of the I-MMSE relation [23] we get

$$\frac{d}{d\Delta^{-1}} \min_{E\in[0,v]} i_{\mathrm{RS}}(E; \Delta) \geq \limsup_{n\to\infty} \frac{1}{n} \frac{dI(\mathbf{S}; \mathbf{W})}{d\Delta^{-1}}. \tag{5}$$

Integrating this relation on $[0, \Delta] \subset [0, \Delta_{\mathrm{AMP}}]$ and checking that $\min_{E\in[0,v]} i_{\mathrm{RS}}(E; 0) = H(S)$ (the Shannon entropy of $P_0$) we obtain $\min_{E\in[0,v]} i_{\mathrm{RS}}(E; \Delta) \leq \liminf_{n\to\infty} I(\mathbf{S}; \mathbf{W})/n$. But we know $I(\mathbf{S}; \mathbf{W})/n \leq \min_{E\in[0,v]} i_{\mathrm{RS}}(E; \Delta)$ [9], thus we already get theorem 1.1 for $\Delta \leq \Delta_{\mathrm{AMP}}$. We notice that $\Delta_{\mathrm{AMP}} \leq \Delta_{\mathrm{Opt}}$. While this might seem intuitively clear, it follows from $\Delta_{\mathrm{RS}} \geq \Delta_{\mathrm{AMP}}$ (by their definitions) which together with $\Delta_{\mathrm{AMP}} > \Delta_{\mathrm{Opt}}$ would imply from theorem 1.1 that $\lim_{n\to\infty} I(\mathbf{S}; \mathbf{W})/n$ is analytic at $\Delta_{\mathrm{Opt}}$, a contradiction. The next step is to extend theorem 1.1 to the range $[\Delta_{\mathrm{AMP}}, \Delta_{\mathrm{Opt}}]$. Suppose for a moment $\Delta_{\mathrm{RS}} \geq \Delta_{\mathrm{Opt}}$. Then *both* functions on each side of the RS formula are analytic on the whole range $]0, \Delta_{\mathrm{Opt}}[$ and since they are equal for $\Delta \leq \Delta_{\mathrm{AMP}}$, they *must* be equal on their whole analyticity range and by continuity, they must also be equal at $\Delta_{\mathrm{Opt}}$ (that the functions are continuous follows from independent arguments on the existence of the $n \to \infty$ limit of concave functions). It remains to show that $\Delta_{\mathrm{RS}} \in ]\Delta_{\mathrm{AMP}}, \Delta_{\mathrm{Opt}}[$ is impossible. We proceed by contradiction, so suppose this is true. Then both functions on each side of the RS formula are analytic on $]0, \Delta_{\mathrm{RS}}[$ and since they are equal for $]0, \Delta_{\mathrm{AMP}}[ \subset ]0, \Delta_{\mathrm{RS}}[$ they must be equal on the whole range $]0, \Delta_{\mathrm{RS}}[$ and also at $\Delta_{\mathrm{RS}}$ by continuity. For $\Delta > \Delta_{\mathrm{RS}}$ the fixed point of SE is $E^\infty = E_{\mathrm{bad}}(\Delta)$ which is also the global minimum of $i_{\mathrm{RS}}(E; \Delta)$, hence (5) is verified. Integrating this inequality on $]\Delta_{\mathrm{RS}}, \Delta[ \subset ]\Delta_{\mathrm{RS}}, \Delta_{\mathrm{Opt}}[$ and using $I(\mathbf{S}; \mathbf{W})/n \leq \min_{E\in[0,v]} i_{\mathrm{RS}}(E; \Delta)$ again, we find that the RS formula holds for all $\Delta \in [0, \Delta_{\mathrm{Opt}}]$. But this implies that $\min_{E\in[0,v]} i_{\mathrm{RS}}(E; \Delta)$ is analytic at $\Delta_{\mathrm{RS}}$, a contradiction.

*We now prove the RS formula for $\Delta \geq \Delta_{\mathrm{Opt}}$.* Note that the previous arguments showed that necessarily $\Delta_{\mathrm{Opt}} \leq \Delta_{\mathrm{RS}}$. Thus by lemmas 3.1 and 3.2 (and the sub-optimality of AMP as shown as before) we obtain $\Delta_{\mathrm{RS}} \leq \Delta_{\mathrm{AMP,coup}} \leq \Delta_{\mathrm{Opt,coup}} = \Delta_{\mathrm{Opt}} \leq \Delta_{\mathrm{RS}}$. This shows that $\Delta_{\mathrm{Opt}} = \Delta_{\mathrm{RS}}$ (this is the point where spatial coupling came in the game and we do not know of other means to prove such an equality). For $\Delta > \Delta_{\mathrm{RS}}$ we have $E^\infty = E_{\mathrm{bad}}(\Delta)$ which is the global minimum of $i_{\mathrm{RS}}(E; \Delta)$. Therefore we again have (5) in this range and the proof can be completed by using once more the integration argument, this time over the range $[\Delta_{\mathrm{RS}}, \Delta] = [\Delta_{\mathrm{Opt}}, \Delta]$.

**Proof sketch of corollaries 1.3 and 1.5:** Let $E_*(\Delta) = \mathrm{argmin}_E i_{\mathrm{RS}}(E; \Delta)$ for $\Delta \neq \Delta_{\mathrm{RS}}$. By explicit calculation one checks that $di_{\mathrm{RS}}(E_*, \Delta)/d\Delta^{-1} = (v^2 - (v - E_*(\Delta))^2)/4$, so from theorem 1.1 and the matrix form of the I-MMSE relation we find $\mathrm{Mmmse}_n \to v^2 - (v - E_*(\Delta))^2$ as $n \to \infty$ which is the first part of the statement of corollary 1.3. Let us now turn to corollary 1.5. For $n \to \infty$ the vector-MSE of the AMP estimator at time $t$ equals $E^t$, and since the fixed point equation corresponding to SE is precisely the stationarity equation for $i_{\mathrm{RS}}(E; \Delta)$, we conclude that for $\Delta \notin [\Delta_{\mathrm{AMP}}, \Delta_{\mathrm{RS}}]$ we must have $E^\infty = E_*(\Delta)$. It remains to prove that $E_*(\Delta) = \lim_{n\to\infty} \mathrm{Vmmse}_n(\Delta)$ at least for $\Delta \notin [\Delta_{\mathrm{AMP}}, \Delta_{\mathrm{RS}}]$ (we believe this is in fact true for all $\Delta$). This will settle the second part of corollary 1.3 as well as 1.5. Using (Nishimori) identities $\mathbb{E}_{\mathbf{S},\mathbf{W}}[S_i S_j \mathbb{E}[X_i X_j | \mathbf{W}]] = \mathbb{E}_{\mathbf{S},\mathbf{W}}[\mathbb{E}[X_i X_j | \mathbf{W}]^2]$ (see e.g. [9]) and using the law of large numbers we can show $\lim_{n\to\infty} \mathrm{Mmmse}_n \leq \lim_{n\to\infty}(v^2 - (v - \mathrm{Vmmse}_n(\Delta))^2)$. Concentration techniques similar to [13] suggest that the equality in fact holds (for $\Delta \neq \Delta_{\mathrm{RS}}$) but there are technicalities that prevent us from completing the proof of equality. However it is interesting to note that this equality would imply $E_*(\Delta) = \lim_{n\to\infty} \mathrm{Vmmse}_n(\Delta)$ for all $\Delta \neq \Delta_{\mathrm{RS}}$. Nevertheless, another argument can be used when AMP is optimal. On one hand the right hand side of the inequality is necessarily smaller than $v^2 - (v - E^\infty)^2$. On the other hand the left hand side of the inequality is equal to $v^2 - (v - E_*(\Delta))^2$. Since $E_*(\Delta) = E^\infty$ when $\Delta \notin [\Delta_{\mathrm{AMP}}, \Delta_{\mathrm{RS}}]$, we can conclude $\lim_{n\to\infty} \mathrm{Vmmse}_n(\Delta) = \mathrm{argmin}_E i_{\mathrm{RS}}(E; \Delta)$ for this range of $\Delta$.

**Proof sketch of lemma 3.1:** Here we prove the lemma for a ring that is not seeded. An easy argument shows that a seed of size $w$ does not change the MI per variable when $L \to \infty$. The statistical physics formulation is convenient: up to the trivial additive term $n(L+1)v^2/4$, the MI $I_{w,L}(\mathbf{S}; \mathbf{W})$ equals the free energy $-\mathbb{E}_{\mathbf{S},\mathbf{Z}}[\ln \mathcal{Z}_{w,L}]$, where $\mathcal{Z}_{w,L} := \int d\mathbf{x} P_0(\mathbf{x}) \exp(-\mathcal{H}(\mathbf{x}, \mathbf{z}, \mathbf{\Lambda}))$ and

$$\mathcal{H}(\mathbf{x}, \mathbf{z}, \mathbf{\Lambda}) = \frac{1}{\Delta} \sum_{\mu=0}^{L} \Big( \Lambda_{\mu\mu} \sum_{i_\mu \leq j_\mu} A_{i_\mu j_\mu}(\mathbf{x}, \mathbf{z}, \mathbf{\Lambda}) + \sum_{\nu=\mu+1}^{\mu+w} \Lambda_{\mu\nu} \sum_{i_\mu, j_\nu} A_{i_\mu j_\nu}(\mathbf{x}, \mathbf{z}, \mathbf{\Lambda}) \Big), \tag{6}$$

with $A_{i_\mu j_\nu}(\mathbf{x}, \mathbf{z}, \mathbf{\Lambda}) := (x_{i_\mu}^2 x_{j_\nu}^2)/(2n) - (s_{i_\mu} s_{j_\nu} x_{i_\mu} x_{j_\nu})/n - (x_{i_\mu} x_{j_\nu} z_{i_\mu j_\nu} \sqrt{\Delta})/\sqrt{n\Lambda_{\mu\nu}}$. Consider a pair of systems with coupling matrices $\mathbf{\Lambda}$ and $\mathbf{\Lambda}'$ and i.i.d noize realizations $\mathbf{z}, \mathbf{z}'$, an *interpolated Hamiltonian* $\mathcal{H}(\mathbf{x}, \mathbf{z}, t\mathbf{\Lambda}) + \mathcal{H}(\mathbf{x}, \mathbf{z}', (1-t)\mathbf{\Lambda}')$, $t \in [0, 1]$, and the corresponding partition function $\mathcal{Z}_t$. The main idea of the proof is to show that for suitable choices of matrices, $-\frac{d}{dt}\mathbb{E}_{\mathbf{S}, \mathbf{Z}, \mathbf{Z}'}[\ln \mathcal{Z}_t] \leq 0$ for all $t \in [0, 1]$ (up to negligible terms), so that by the fundamental theorem of calculus, we get a comparison between the free energies of $\mathcal{H}(\mathbf{x}, \mathbf{z}, \mathbf{\Lambda})$ and $\mathcal{H}(\mathbf{x}, \mathbf{z}', \mathbf{\Lambda}')$. Performing the $t$-derivative brings down a Gibbs average of a polynomial in all variables $s_{i_\mu}, x_{i_\mu}, z_{i_\mu j_\nu}$ and $z'_{i_\mu j_\nu}$. This expectation over $\mathbf{S}, \mathbf{Z}$, $\mathbf{Z}'$ of this Gibbs average is simplified using integration by parts over the Gaussian noise $z_{i_\mu j_\nu}, z'_{i_\mu j_\nu}$ and Nishimori identities (see e.g. proof of corollary 1.3 for one of them). This algebra leads to

$$-\frac{1}{n(L+1)}\frac{d}{dt}\mathbb{E}_{\mathbf{S}, \mathbf{Z}, \mathbf{Z}'}[\ln \mathcal{Z}_t] = \frac{1}{4\Delta(L+1)}\mathbb{E}_{\mathbf{S}, \mathbf{Z}, \mathbf{Z}'}[\langle \mathbf{q}^\mathsf{T}\mathbf{\Lambda}\mathbf{q} - \mathbf{q}^\mathsf{T}\mathbf{\Lambda}'\mathbf{q}\rangle_t] + \mathcal{O}(1/(nL)), \quad (7)$$

where $\langle - \rangle_t$ is the Gibbs average w.r.t the interpolated Hamiltonian, $\mathbf{q}$ is the vector of overlaps $q_\mu := \sum_{i_\mu=1}^n s_{i_\mu} x_{i_\mu}/n$. If we can choose matrices s.t $\mathbf{\Lambda}' > \mathbf{\Lambda}$, the difference of quadratic forms in the Gibbs bracket is negative and we obtain an inequality in the large size limit. We use this scheme to interpolate between the fully decoupled system $w = 0$ and the coupled one $1 \leq w < L/2$ and then between $1 \leq w < L/2$ and the fully connected system $w = L/2$. The $w = 0$ system has $\Lambda_{\mu\nu} = \delta_{\mu\nu}$ with eigenvalues $(1, 1, \ldots, 1)$. For the $1 \leq w < L/2$ system, we take any stochastic translation invariant matrix with non-negative discrete Fourier transform (of its rows): such matrices have an eigenvalue equal to $1$ and all others in $[0, 1[$ (the eigenvalues are precisely equal to the discrete Fourier transform). For $w = L/2$ we choose $\Lambda_{\mu\nu} = 1/(L+1)$ which is a projector with eigenvalues $(0, 0, \ldots, 1)$. With these choices we deduce that the free energies and MIs are ordered as $I_{w=0,L} + \mathcal{O}(1) \leq I_{w,L} + \mathcal{O}(1) \leq I_{w=L/2,L} + \mathcal{O}(1)$. To conclude the proof we divide by $n(L+1)$ and note that the limits of the leftmost and rightmost MIs are equal, provided the limit exists. Indeed the leftmost term equals $L$ times $I(\mathbf{S}; \mathbf{W})$ and the rightmost term is the *same* MI for a system of $n(L+1)$ variables. Existence of the limit follows by subadditivity, proven by a similar interpolation [18].

**Proof sketch of lemma 3.2:** Fix $\Delta < \Delta_{\mathrm{RS}}$. We show that, for $w$ large enough, the coupled SE recursion (4) must converge to a fixed point $E_\mu^\infty \leq E_{\mathrm{good}}(\Delta)$ for all $\mu$. The main intuition behind the proof is to use a "potential function" whose "energy" can be lowered by small perturbation of a fixed point that *would* go above $E_{\mathrm{good}}(\Delta)$ [16, 17]. The relevant potential function $i_{w,L}(\mathbf{E}, \Delta)$ is in fact the replica potential of the coupled system (a generalization of (1)). The stationary condition for this potential is precisely (4) (without the seeding condition). Monotonicity properties of SE ensure that any fixed point has a "unimodal" shape (and recall that it vanishes for $\mu \in \mathcal{B} = \{0, \ldots, w-1\} \cup \{L-w, \ldots, L\}$). Consider a position $\mu_{\max} \in \{w, \ldots, L-w-1\}$ where it is maximal and suppose that $E_{\mu_{\max}}^\infty > E_{\mathrm{good}}(\Delta)$. We associate to the *fixed point* $\mathbf{E}^\infty$ a so-called *saturated profile* $\mathbf{E}^{\mathrm{s}}$ defined on the whole of $\mathbb{Z}$ as follows: $E_\mu^{\mathrm{s}} = E_{\mathrm{good}}(\Delta)$ for all $\mu \leq \mu_\infty$ where $\mu_\infty + 1$ is the smallest position s.t $E_\mu^\infty > E_{\mathrm{good}}(\Delta)$; $E_\mu^{\mathrm{s}} = E_\mu^\infty$ for $\mu \in \{\mu_\infty + 1, \ldots, \mu_{\max}-1\}$; $E_\mu^{\mathrm{s}} = E_{\mu_{\max}}^\infty$ for all $\mu \geq \mu_{\max}$. We show that $\mathbf{E}^{\mathrm{s}}$ cannot exist for $w$ large enough. To this end define a shift operator by $[\mathcal{S}(\mathbf{E}^{\mathrm{s}})]_\mu := E_{\mu-1}^{\mathrm{s}}$. On one hand the shifted profile is a *small* perturbation of $\mathbf{E}^{\mathrm{s}}$ which matches a fixed point, except where it is constant, so if we Taylor expand, the first order vanishes and the second order and higher orders can be estimated as $|i_{w,L}(\mathcal{S}(\mathbf{E}^{\mathrm{s}}); \Delta) - i_{w,L}(\mathbf{E}^{\mathrm{s}}; \Delta)| = \mathcal{O}(1/w)$ uniformly in $L$. On the other hand, by explicit cancellation of telescopic sums $i_{w,L}(\mathcal{S}(\mathbf{E}^{\mathrm{s}}); \Delta) - i_{w,L}(\mathbf{E}^{\mathrm{s}}; \Delta) = i_{\mathrm{RS}}(E_{\mathrm{good}}; \Delta) - i_{\mathrm{RS}}(E_{\mu_{\max}}^\infty; \Delta)$. Now one can show from monotonicity properties of SE that if $\mathbf{E}^\infty$ is a non trivial fixed point of the coupled SE then $E_{\mu_{\max}}^\infty$ *cannot be in the basin of attraction of* $E_{\mathrm{good}}(\Delta)$ for the uncoupled SE recursion. Consequently as can be seen on the plot of $i_{\mathrm{RS}}(E; \Delta)$ (e.g. figure 1) we must have $i_{\mathrm{RS}}(E_{\mu_{\max}}^\infty; \Delta) \geq i_{\mathrm{RS}}(E_{\mathrm{bad}}; \Delta)$. Therefore $i_{w,L}(\mathcal{S}(\mathbf{E}^{\mathrm{s}}); \Delta) - i_{w,L}(\mathbf{E}^{\mathrm{s}}; \Delta) \leq -|i_{\mathrm{RS}}(E_{\mathrm{bad}}; \Delta) - i_{\mathrm{RS}}(E_{\mathrm{good}}; \Delta)|$ which is an energy gain independent of $w$, and for large enough $w$ we get a contradiction with the previous estimate coming from the Taylor expansion.

## Acknowledgments

J.B and M.D acknowledge funding from the SNSF (grant 200021-156672). Part of this research received funding from the ERC under the EU's 7th Framework Programme (FP/2007-2013/ERC Grant Agreement 307087-SPARCS). F.K and L.Z thank the Simons Institute for its hospitality.

## Footnotes

[1]We take $\mathbb{E}[S] \neq 0$. Once theorem 1.1 is proven for this case a limiting argument allows to extend it to $\mathbb{E}[S] = 0$.

[2] Note that here since $E = v = 1$ is an extremum of $i_{\mathrm{RS}}(E; \Delta)$, one must introduce a small bias in $P_0$ and let it then tend to zero at the end of the proofs.

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
