[Reviews · NeurIPS 2016]

Reviewer 1

Summary

The authors consider the rank-one matrix estimation problem and aim to understand precisely the mutual information between the observed noisy matrix W and the original rank-one matrix SS^T. They show (under certain assumptions) that the asymptotic mutual information per variable is given by the minimum of the so-called replica-symmetric potential function. This then gives a mean to compute the information-theoretic threshold for reconstructing the signal S. The results imply that approximate message-passing (AMP) achieves the MMSE when the noise is either small or large. In between the two there can be a very interesting regime where AMP is not optimal, and there seems to be a computational gap. The authors illustrate their results nicely on two models: the Wigner spike model and community detection. The proof involves a very nice combination of techniques from different areas. The answer was predicted using statistical physics techniques, and so it is natural that these appear in the proof; in particular the interpolation method introduced by Guerra is used. This is combined with recent progress in coding theory on capacity-achieving spatially coupled codes.

Qualitative Assessment

The paper is very well written. It presents a very nice proof of a prediction from statistical physics about the mutual information for symmetric rank-one matrix estimation. The proof techniques have the potential to be applied more widely in order to obtain a better rigorous understanding about the fundamental limits of low-rank matrix estimation. I believe that this paper is a valuable contribution to NIPS. My main comment is that the proof sketches are dense; but this is unavoidable for a conference version. I therefore urge the authors to prepare a more detailed journal version of the paper.

Confidence in this Review

2-Confident (read it all; understood it all reasonably well)


Reviewer 2

Summary

The paper proves a conjectured formula of mutual information for symmetric rank-one matrix estimation under the AWGN model. It shows that the limit of mutual information can be expressed as the minimum of the replica symmetric potential function for a wide range of signal distributions, which also gives an exact formula for the MMSE for reconstructing the signal. The result identifies an interesting regime where the information-theoretic optimal performance is not achieved by AMP. The implication of the result is further illustrated by two examples of Wigner spike model and community detection. The proof appears to be non-trivial and novel.

Qualitative Assessment

The paper is a very nice piece of theoretic work and quite a fun to read. The two sets of figures and the two illustrating examples are particularly entertaining and enlightening. Although similar results have been proved in other work under various models, the current paper seems to be the first to prove the conjectured formula in its full generality and is able to identify a gap between information theoretic lower bounds and algorithmic upper bounds. The proofs appear to be non-trivial and could possibly shed lights on other problems. I recommend the paper to be accepted by NIPS. Some minor suggestion: - It'd be nice if some background and reference are provided for the replica symmetric potential function for better readability. - Publication year seems to be missing for some of the references: [3], [7], [20] - All "pca"'s appear in lower cases in the references.

Confidence in this Review

1-Less confident (might not have understood significant parts)


Reviewer 3

Summary

This paper proves, in a rank-one case, a conjectured general expression for the mutual information in the problem of factoring low-rank matrices from noisy observations. The key ideas are to use state evolution to analyse behaviors of approximate message passing (AMP) applied to the problem, and to apply the idea of spatial coupling in coding theory in order to make AMP performing well beyond the thredhold at which the conventional AMP fails.

Qualitative Assessment

Even though the arugment in this paper is limited to a rank-one case, it nevertheless provides results covering various problems in application domains, such as the spiked Wigner model and community detection in stochastic block model. I think that the main problem with this paper is that it is very dense in showing some details of the proofs of the main results. I feel that it is still the case even though I consider the theoretical nature of this paper. Due to page limitation, description in this paper was forced to be brief, which altogether makes the paper hard to follow. It would have been better if some parts of the description of the proofs are moved in supplementary material to put more space to the discussion section, in order to appeal a wider audience of the conference. Lines 153-154: I do not understand what is meant here. Is the analysis valid in any arbitrary scaling of \rho in n? If so, then it should be stated explicitly, but it seems that \rho=O(1) is assumed in the analysis. In the community detection problem, I wonder whether the assertion \Delta_Opt=1 for \rho>\rho_c is valid in view of Definition 1.2, as well as whether \Delta_AMP for the same range of \rho is valid in terms of Definition 1.4. It seems that for \rho>\rho_c, at \Delta=1 a second-order transition occurs, so that although the former assertion would be fine the latter would not be justifiable. In the insets of Figure 2, it seems as if MSE for MMSE, as a function of \Delta, has an infinite derivative at \Delta_Opt, which would seem inconsistent with the Maxwell rule. Lines 227-8: The term matrix-MMSE is used without proper definition. I do not understand what is meant by the sentence "This cannot improve the matrix-MMSE." In the reference list, some items (references 2, 3, 7, 8, 14, 18, and 23, as far as I noticed) have incomplete bibliographic information.

Confidence in this Review

2-Confident (read it all; understood it all reasonably well)


Reviewer 4

Summary

The paper considers the estimation of a vector S by observing noisy versions W_{ij} of its coordinate-wise product terms s_i.s_j normalized by \sqrt{n}, under additive Gaussian noise of variance \Delta. The main result is a formula for the limit (1) \lim_{n --> \infty} 1/n I(S; W) in terms of the replica symmetric potential function i_{RS}(E, \Delta), which, as a corollary (using the I-MMSE relation), shows that MMSE(\Delta) for the estimation problem above is given by the minimizer E^* of i_{RS}(E, \Delta). As a further corollary it is shown that Approximate Message Passing (AMP), which is known to converge to a stationary point of i_{RS}(E, \Delta) when \Delta is in appropriate range, yields MMSE in the same range. The proof of (1) is the main contribution of the paper. The proof is based on establishing an equality of the two regions over an appropriate interval and then extending it to a larger region over which the (complex extensions of the) two sides are analytic. A key step is to show that the analytic threshold for the left-side of (1), termed \Delta_{OPT}, is at least as much as that for the right-side, \Delta_{RS}. This is facilitated by relating the two thresholds to the threshold for a spatially coupled construction of W_{ij} (which is done in Lemma 3.1 and Lemma 3.2).

Qualitative Assessment

The paper addresses an important topic, namely that of providing formal guarantees for algorithms derived from heuristics in statistical physics. The approach proposed in the paper is very interesting and perhaps novel, at to this reviewer it is, wherein the gap in the thresholds for information theoretic performance and the performance of AMP is bridged by connecting the information theoretic threshold and that for AMP to that of a spatially coupled model. This relation in turn relies on the equality of the threshold for spatially coupled construction and the optimal information theoretic threshold, a phenomenon that has led to a recent breakthrough capacity achieving construction in information theory and is proved here again via Lemma 3.1. Another key step is to show that \Delta_{RS} does not exceed the threshold for the spatially coupled construction with AMP; this is the content of Lemma 3.2. While only a sketch of proof is presented, the approach looks convincing and indeed very interesting. The analysis proposed in the paper, if rigorously correct, leads to an interest approach for analyzing the performance of AMP by bringing in a spatially coupled system to relate various thresholds. Whether this contribution is new is beyond the knowledge of this reviewer. Nevertheless, this overall approach is very fascinating. In addition to the general comments above, I have the following specific comments: 1. The paper and the contribution is rather difficult to follow. The authors mention the connection of their results to several problems but then only present two applications via examples in Section 2 without clarifying the exact implication of their approach for these examples. In particular, what was the state-of-the-art before this paper and how has this paper furthered the analysis of these applications? Also, I suggest that the authors spend some more space in explaining the connection between the formula derived here and the general inference problem. Perhaps the relation of the left-side to information theoretic threshold and the right-side to the threshold of AMP (via the replica symmetric potential function) can be mentioned at the outset without much details. 2. The proof sketch is very difficult to follow. The authors should first summarize the plan before getting to the details. I have written above what I could follow, which might be wrong. However, a similar description in words of the proof idea of the authors will be very useful. 3. The sentence leading to (8) and the one following it are unclear to me. In particular, I couldn't follow the observation "so the left hand side of (7) is equal to the derivative of \min_{E}i_{RS(E, \Delta) w.r.t. \Delta^{-1}". Does the other term in i_{RS}(E, \Delta) equals 0 at the min? Also, in the sentence after (8), it seems you interchanged the limit and the integral. If you justify this interchange in a complete version, at least point-out in the sketch that it needs to be done. Finally, understanding the proof sketch of Lemma 3.2 seems beyond the scope of a non-expert. 4. The extensions claimed in 135-39 seem to be much more interesting than the simple case here. If the authors indeed have these extensions available, why not motivate the paper by these extended results. If such extensions are not available as of now, please modify the claim above to be clear about it.

Confidence in this Review

2-Confident (read it all; understood it all reasonably well)


Reviewer 5

Summary

This paper tries to give a rigorous proof of the expression for the mutual information in the symmetric rank-one matrix model, instead of the statistical physics one, which is considered not rigorous. It first presents the model in claiming that in most case, the asymptotic mutual information can be approximated by the one in the additive white Gaussian noise (AWGN) setting. Then, it gives a list of interleaved formula, hypotheses, definitions and theorems, without explaining them in detail. Next, it begins to interpret the gap between the information theoretical and algorithmic AMP threshold, by explaining how the phase transition happens when the level of noise increases. Then, in Section 2, the paper gives two use cases: Wigner spike model and community detection. Finally, in Section 3, the paper provides a sketch of proof.

Qualitative Assessment

This paper is written in a casual style. A lot of definitions are missing, the formula are poorly explained, the description is ambiguous and the proofs are extremely concise. It is impossible to understand unless the reader is currently working on the same problem. The reviewer comes from the statistical learning community, who does not have much experience on information theory. With great patience and persistence, and numerous hours spent reading this paper, he gives the following remarks. Bayes optimal. In many cases, the terms "Bayes" and "optimal" can mean many different things. The reader may once have learned the "Bayes optimal classifier" during the school, but he may well have forgotten it. Anyway, it cannot be called a “setting” at L2 in the abstract (zero hit on Google). It had better describe it seriously in the main part. Heuristic statistical physics computation (zero hit on Google). In my area, people call an algorithm heuristic if it hasn't been proven to converge to something desired. In this paper, instead, it is used to describe some calculation without sound mathematical meaning, such as casually interchanging the order of integral, writing some integrals without mathematical definition (e.g., the epoch when Lebesgue integral hasn't be invented). Better clarify this point in the main part. MSE. Mean square error can have many kinds of formulation. Theoretically, from any two random variables, one can calculate an MSE. Better give a clear formula. AMP. Approximate message passing being the most important component of this paper, however, one cannot find the definition of this algorithm in this paper. L8: "a large set of parameters". There are lots of parameters in the world: parameters for models, for algorithms, hyper-parameters... Should be "a large extent of level/degree of noise". Channel universality (L40). The paper cites [9], suggesting that it is proven in [9]. Then why does it continue to cite [4] and [10], leaving the impression that it hasn't been completely proven? By the way, [4] and [9] are on arXiv, which means they may not have gone through serious peer review. AWGN. This abbrev has never been properly introduced, though the reader can guess it is "additive white Gaussian noise". Eq (3). Should explain that I(S; W) is a uni-variable function of the level of noise --Delta, and should emphasize that the Delta in i_RS(E; Delta) is the same Delta as the level of noise, because the RSPF can happily exist alone without making any connection with the rest of the world. A rigorous reader may well claim that there are two different Delta's in Eq (3), and thus disqualify this paper. Eg (2). Should give the interpretation of E (error), otherwise will be difficult to understand the rest of the paper. Community detection. Not clear in this paper how it is linked with the symmetric rank-one matrix estimation. Figure 1. A conflict in the caption. The paper reads "... (1), (2) and (4), but in (3), when Delta_AMP < Delta < Delta_RS, ...". However, (3) is for case "Delta = 0.00125 > Delta_RS". This conflict may be due to inconsistent read of the figure: sometimes the author read it from up to down, other times he read it from left to right. The reviewer suggests adding the No. directly onto each sub-figure. Contribution. It is not clear in the paper whether the threshold gap is one of the contribution of this article.

Confidence in this Review

1-Less confident (might not have understood significant parts)


Reviewer 6

Summary

This paper focuses on rank-one matrix estimation problem. Suppose we have a noisy observation of rank one matrix W = ss^T + \Delta, and we want to reconstruct s. The main contribution of the paper is to identify the asymptotic mutual information I(S;W) via replica symmetric potential function i_{RS}. The relation of approximate message passing algorithm and the quantity i_{RS} is also studied in this paper.

Qualitative Assessment

Pros: 1. The authors have good mathematical ability to prove theorems. Cons: 1. I don't understand the meaning of replica symmetric potential function i_{RS} (which plays the central role in this paper). The formula (2) is too complicated to understand. So it makes me unable to understand the follow-on results. Maybe it is good to explain i_{RS} after giving the formula. 2. I don't understand how to use the main result in practice. It seems that there are no practical algorithm presented or any guide to use the theorem to improve or explain some existing algorithms.

Confidence in this Review

1-Less confident (might not have understood significant parts)